# Markers for the Prediction of Probably Sarcopenia in Middle-Aged Individuals

**DOI:** 10.3390/jpm12111830

**Published:** 2022-11-03

**Authors:** Yulia G. Samoilova, Mariia V. Matveeva, Ekaterina A. Khoroshunova, Dmitry A. Kudlay, Oxana A. Oleynik, Liudmila V. Spirina

**Affiliations:** 1Federal State Budgetary Educational Institution of Higher Education «Siberian State Medical University» of the Ministry of Health of Russia, Moskovsky Trakt 2, 634050 Tomsk, Russia; 2Federal State Autonomous Educational Institution of Higher Education “First Moscow State Medical University Named after I.I. THEM. Sechenov” of the Ministry of Health of Russia (Sechenov University), St. Trubetskaya 8, Building 2, 119048 Moscow, Russia; 3Federal State Budgetary Institution “State Research Center “Institute of Immunology”” FMBA of Russia, Kashirskoe sh., 24, 115478 Moscow, Russia

**Keywords:** pro-inflammatory markers, probably sarcopenia, middle-aged individuals

## Abstract

Sarcopenia is a condition that is characterized by a progressive loss of muscle mass, strength, and function, resulting in reduced quality of life. The aim of the study was to analyze the significance of pro-inflammatory markers in the prognostic diagnosis of sarcopenia. The participants were divided into two groups: the main group of 146 people and the control—75 people. The complex of examinations included neuropsychological testing (Hospital Anxiety and Depression Scale (HADS), quality-of-life questionnaire for patients with sarcopenia (SarQoL), and short health assessment form (MOS SF-36)), a 6 m walking speed test, manual dynamometry, bioimpedancemetry, and metabolic markers (nitrates, fibroblast growth factor 21, and malondialdehyde). When analyzing metabolic markers in the main group, a twofold increase in nitrates in the main group was recorded in a subsequent analysis adjusted for multiple variables, there was a negative association between the nitrate levels for weak grip strength and appendicular muscle mass. An additional analysis revealed that the complaint of pain in the lower extremities was more frequent in patients of the main group, as well as constipation and the pathology of thyroid gland, and they were more frequently diagnosed with arterial hypertension. At the same time, patients from the main group more frequently took vitamin D. When conducting body composition, the main group recorded a higher weight visceral fat content, as well as a decrease in appendicular and skeletal muscle mass; these changes were accompanied by a decrease in protein and minerals. Among the markers that differed significantly were nitrates, and it was this that was associated with decreased muscle strength and appendicular mass, which may indicate both a possible mechanism and a possible predictive marker. The results of this study can be used to develop a screening method for diagnosing sarcopenia at the outpatient stage.

## 1. Introduction

Every year, the number of people in the world increases by 250 thousand people, mainly due to the elderly [1]. According to experts of the United Nations Organization, the number of the elderly will increase to 21% by 2050 and will amount to 2.1 billion people [2]. One condition that worsens prognosis and quality is sarcopenia. Sarcopenia is a progressive and generalized skeletal muscle disorder involving the accelerated loss of muscle mass and function that is associated with increased adverse outcomes, including falls, functional decline, frailty, and mortality. It occurs commonly as an age-related process in older people, influenced not only by contemporaneous risk factors, but also by genetic and lifestyle factors operating across the life course. It can also occur in mid-life in association with a range of conditions. The prevalence of sarcopenia in persons 60–70 years old is 5–13%, and in the group over 80 years old, it increases to 50% [3]. Currently, the number of patients with sarcopenia in the world is 50 million people, and this figure is predicted to quadruple in 40 years [4]. According to studies, presarcopenia is more susceptible to men over the age of 60 years and women over 80 years old [5]. The prevalence of sarcopenia is 10–15% lower than in Russia [6].

The pathogenesis of sarcopenia is quite complex; at present, many factors have been identified that affect the decrease in muscle mass, among which endocrine ones play a significant role. At the same time, the prognosis remains unfavorable since there are no registered pathogenetic drugs for the treatment of sarcopenia [7].

Severe sarcopenia is known to be associated with decreased quality of life and symptoms of depression, which requires mandatory diagnosis in the examination, but there is no information on such changes in probable sarcopenia, which is also interesting to study [8].

As one of the biomarkers, we consider fibroblast growth factor 21 (FGF21), which, in a healthy population, is responsible for many processes, including muscle function [9]. In one study, FGF21 was identified as another independent biomarker of successful aging, increased by a counterregulatory mechanism when compensating for metabolic stress, including in muscle tissue [10]. Circulating levels of FGF21 are elevated in patients with mitochondrial disorders affecting skeletal muscle [11]. Thus, FGF21 may be involved in the regulation of body muscle and in the modulation of metabolism, and thus be associated with presarcopenia.

Other promising metabolites are nitrates. In the human body, nitric oxide (NO) is one of the key endogenous regulators of the cardiovascular and other systems. The endothelium can be considered a giant endocrine organ in which the cells are not collected together, as in the endocrine glands, but dispersed, in the vessels. Its activation is mainly associated with hemodynamic changes occurring in the body [12]. Many articles have been published information about the effect of nitrates on the cardiovascular system. However, there are no large-scale studies in patients with sarcopenia. Nitric oxide synthase inhibitors (iNOS) have been studied as a therapeutic point [13]. This scientific work revealed a mutual correlation of high iNOS expression with activation of cytokine-induced storm. Inhibitors of iNOS activity had a positive effect on the prevention of muscle mass reduction [14]. The main consequence of the decrease in NO in the muscle is a decrease in the flow of vasodilation, which directly correlates with muscle strength and physical functioning, respectively, can influence the development of sarcopenia [15].

Malonic dialdehyde is a produced form of peroxidation of polyunsaturated fatty acids. This marker easily reacts with proteins, forming a wide range of intra- and intermolecular covalent adducts. These metabolic products gradually accumulate in the vascular system, leading to cellular dysfunction and tissue damage [16]. Loss of muscle mass in healthy older adults is associated with the development of lipid peroxidation, and taking antioxidants or exercising improves muscle-system parameters, indicating a possible evaluation of antioxidant system parameters in the preclinical stage of sarcopenia [17,18].

Currently, there is evidence that the loss of muscle mass occurs much earlier, starting as early as age 55 in men and 45 in women, respectively [19]. The diagnosis of sarcopenia rests on muscle-mass measurements and on functional tests that evaluate either muscle strength or physical performance (walking and balance). No specific biomarkers have been identified to date. The identification of a single metabolic marker or a panel of markers can verify a decrease in muscle mass in the preclinical stage, when preventive and rehabilitative measures are effective. The aim of the study was to analyze the significance of pro-inflammatory markers in the prognostic probably sarcopenia in middle-aged adults.

## 2. Materials and Methods

The study was performed in accordance with Good Clinical Practice and the principles of the Declaration of Helsinki. The study protocol was approved by the ethical committee of Siberian State Medical University No. 8888 dated 29 November 2021. Prior to inclusion in the study, written informed consent was obtained from all participants.

The study was conducted in the medical prevention office of the Center for Public Health and Medical Prevention for one-quarter of 2022. All patients who were 45–85 years old, who have signed informed consents, with no exclusion criteria were included in the study (Figure 1). Of the many criteria, those that could affect the results of the study have been chosen, meaning that it is they, not sarcopenia, that will determine the results obtained even as early as the first stage of randomization, which would violate the purity of the experiment. Vital functions, including blindness, can affect physical activity and muscle mass. In this regard, the status of these functions was necessarily taken into account during randomization. Exclusion criteria: pregnant women, alcohol abusers, patients with liver diseases, respiratory and cardiovascular systems, gastrointestinal tract in decompensation stage, musculoskeletal pathology, moderate-or-severe cognitive impairment, and immobilization in the last 6 months. A total of 460 patients were included in the study. Of all included patients, 158 were excluded due to musculoskeletal pathology, 56 had cognitive impairment of moderate severity or dementia, 15 had blindness, and 16 had stroke. A total of 221 people aged 45 to 85 years participated, including 145 women and 56 men (the predominance of women in the study is associated with more frequent use of medical care for disease prevention). 

The patients were divided into 2 groups according AWGS 2019: the main group with reduced muscle mass of 146 people (118 women and 28 men) and the control group of 75 people (47 women and 28 men) based on testing [20]. A dynamometer and walking-speed test were used as primary testing to differentiate into groups. Patients were divided into groups according to the results of physical strength and function assessment, thus belonging to the group according to the classification features “probable sarcopenia”. The protocol included measuring handgrip strength while taking maximum readings in two trials, using both arms in isometric contraction with maximum force; the standard was sitting position with 90° elbow extension for the dynamometer. The diagnostic criteria for low handgrip muscle strength was <28.0 kg for men and <18.0 kg for women [20]. To assess the functions, we measured the time taken to walk 6 m at a normal pace from the start without slowing down and took the average result of at least two trials, using a handheld stopwatch, as the recorded speed. A level ≤0.8 m/s to <1.0 m/s, independent of floor, was used for the criteria.

A screening test SarQol for probable sarcopenia was performed on everyone, but because the patients included in the pain study were younger than the generally accepted age for sarcopenia, it was not used as a cutoff point. To exclude the influence of neuropsychological parameters on the development of sarcopenia, all patients underwent an assessment. For this purpose, we used the hospital anxiety assessment scale (HADS), as well as the general quality of life questionnaire (SF36 SF36).

All participants underwent bioimpedance measurement with the help of an Inbody 770 device (Korea), with an evaluation of the following parameters: analysis of proteins, minerals, total body water, total weight, fat mass, skeletal muscle mass, muscle mass, analysis of extracellular and intracellular fluid, analysis of fat tissue by segments (right arm, left arm, trunk, right leg, and left leg), and appendicular skeletal mass index. The appendicular skeletal mass index (SMI) was calculated by the ratio of lean muscle mass of all to the height parameter in square meters. Measurements were taken according to the manufacturer’s manual, in a vertical position, with the legs and arms in contact with the base electrodes and the capture electrodes, respectively. AWGS 2019 low muscle mass values of <7.0 kg/m^2^ and <5.7 kg/m^2^ for men and women, respectively, were used as cutoff points [19].

For the study of metabolic markers, blood was taken, with further laboratory analysis: fibroblast growth factor 21, nitrates, malondialdehyde (MDA), and lactate dehydrogenase by enzyme immunoassay.

A statistical analysis and data processing were performed by using SPSS Statistica software. For sample-size calculations, we took limb extremities’ muscle mass as the outcome index and used the mean and standard deviation values in Shahar et al. with a fixed power of 80% and an α level of 5% for the main variable [21]. This gave a sample size of 70 subjects for each group. However, by taking into account the exclusion criteria and possible dropout from the study, the main and control groups were increased in size. The methods of descriptive statistics were quartiles—for non-normally distributed data. Testing of statistical hypotheses of normally distributed quantitative parameters was performed by using the following parametric criteria: Student’s *t*-test for paired comparison. The correlation analysis was assessed by using Pearson’s criterion. To assess reliability of differences in qualitative characteristics, we used conjugation tables with the calculation of χ^2^ (chi-square). The null hypothesis was rejected at the level of statistical significance *p* < 0.05.

## 3. Results

This section presents the results of the examination of patients with probable sarcopenia, who were divided according to gender (because there are gender differences and norms) in the tables.

The characteristics of the patients are presented in Table 1.

We analyzed neuropsychological characteristics, namely depression and quality of life, and the groups did not differ (Table 2).

Table 3 presents data on the composition of the body.

According to the bioimpedancemetry data, the main group showed an increase in fat mass, percentage of fat mass, and visceral fat area compared with the control group, as well as a more pronounced decrease in skeletal muscle mass than in the control group and also content of protein and minerals. The appendicular mass scores were reduced in both men and women in the main group.

The next step was the evaluation of proinflammatory and oxidative stress markers in patients with and without sarcopenia (Table 4).

When analyzing the metabolic parameters in the main group, there was an increase in nitrates in comparison with the control two times (*p* = 0.05). Fibroblast growth factor 21 and malondialdehyde values were not statistically significant.

In a subsequent analysis adjusted for multiple variables, there was a negative association of nitrate levels for weak grip strength (*p* = 0.004) and appendicular muscle mass (*p* = 0.005).

An additional analysis revealed that the complaint of pain in the lower extremities was more frequent in patients of the main group (*p* = 0.023), as well as constipation (*p* = 0.008) and pathology of thyroid gland (*p* = 0.026), and they were more frequently diagnosed with arterial hypertension (*p* = 0.032). At the same time, patients from the main group more frequently took vitamin D (*p* = 0.014).

So, as a result of the study, it was found that the groups did not differ according to the questionnaires. The dynamometry values for the main group were lower than those for the control group. The walking speed test did not allow us to objectively evaluate the functional ability of the muscles, because the walking time in the control group was longer than in the main group, both in men and in women. When conducting body composition, the main group recorded a higher weight and visceral fat content, as well as a decrease in appendicular and skeletal muscle mass, these changes were accompanied by a decrease in protein and minerals. Among the markers that differed significantly were nitrates, and it was this that was associated with decreased muscle strength and appendicular mass, which may indicate both a possible mechanism and a possible predictive marker, as well as malonic dialdehyde indices, which are significantly increased in the main group, but are not statistically significant.

## 4. Discussion

There is now evidence that sarcopenia can be diagnosed at a younger age, so in Poland, sarcopenia has been found in 4.5–5.1% at a young age (20–35 years) by appendicular mass [22]. In this regard, the relevance of assessing the signs or predicting factors for the development of sarcopenia at an age less than 65 years is of interest and will also allow the preclinical stage to identify the risk group and conduct rehabilitative measures.

Screening by questionnaires for depression and quality of life in patients with sarcopenia did not differ between the groups. This is probably because the participants were middle-aged patients who did not have significant clinical symptoms of sarcopenia yet. In a recent study the investigators assessed quality of life in patients with osteoporosis/osteopenia and sarcopenia, and the SF-36 questionnaire showed decreased quality of life; it should be considered that this study involved people over 65 years of age [23]. In a survey of elderly people aged 65 years and older living in South Korea, the relationship between reduced quality of life by the SarQoL questionnaire and nutritional risk was statistically significant [24]. Therefore, these questionnaires have little significance in persons with preclinical signs of sarcopenia, but they can be used in the older age group to assess disease progression and therapy efficacy.

Perhaps a more extensive questionnaire than the SARC-F is needed to assess nutritional status. For example, a study was conducted in Poland to evaluate the effectiveness of a version of the MRSA (Mini Sarcopenia Risk Assessment) questionnaire. It includes questions assessing age, weight loss in the last year, frequency of hospitalizations in the last year, physical activity level, frequency of food intake, and intake of dairy products and protein. As a result, PL-MSRA-5 is more effective than PL-MSRA-7 for sarcopenia risk assessment [25]. An analysis of a patient’s nutritional status is very important because a lack of protein-containing foods leads to a progressive decline in muscle mass [26].

Although the sarcopenia screening questionnaire did not differ between groups and showed no risk of sarcopenia, muscle mass and muscle function were reduced. The diagnosis was confirmed by bioimpedanceometry; in middle-aged patients, in addition to changes in adipose tissue, altered skeletal muscle was registered, and the diagnosis of sarcopenia was confirmed by sarcopenia. Sarcopenic obesity is associated with low functional status and high mortality. The low prevalence of verification of the diagnosis seems to be due to the underestimation of sarcopenia in obese individuals; this can be avoided when using the body-mass-index assessment, where an isolated skeletal musculature assessment is performed, and the appendicular skeletal mass index. It should be noted that this condition occurs not only in middle age but also at a young age. The Danish study assessing sarcopenic obesity involved a group aged 20 to 59 years with a BMI ≥ 35 kg/m^2^ and comorbidities or a BMI > 40 kg/m^2^. The body composition was assessed by dual-energy X-ray absorptiometry (DEXA), and muscle function and strength were assessed by dynamometry and a functional test (standing up from a chair five times). As a result, low leg-muscle strength was found in 33% of participants, and the prevalence of sarcopenia ranged from 11.1% to 13.9%, with a higher prevalence in middle-aged women than in men [27]. Our study confirms these findings and, once again, proves the importance and timeliness of diagnosing a decrease in muscle mass and strength in middle-aged individuals.

Muscle mass showed a moderate correlation with hand-grip strength, which is partly explained by age-related fibrosis of some myocytes, as well as their replacement by adipose tissue [28]. These data are supported by high levels of fat mass in our study. Studies show that the degree of obesity is associated with greater muscle mass and absolute muscle strength when compared to non-obese people. However, when the body mass or muscle mass are normalized, people living with obesity have reduced muscle performance in comparison with individuals of normal weight. Therefore, a high-protein diet, in combination with exercise, should be used when losing weight in order to maintain lean body mass and prevent loss of muscle mass in the process of weight loss [29].

The muscle-weakness criterion and hand dynamometry scores of the EWGSOP2 algorithms are likely to identify different populations of older people as likely to have sarcopenia, given the same prevalence when either measure was used alone, but when both measures were used simultaneously, the prevalence of sarcopenia was much higher. The decrease in bone mass in the main group can be explained by an excessive amount of visceral fat, which is associated with increased proinflammatory activity of adipocytes and insulin resistance, which eventually leads to changes in the protein structure of the bones [30]. The results of our study contradict data obtained by colleagues in Spain, where the combined prevalence of sarcopenic obesity and osteoporosis was 0% [31]. This is due to the fact that, in this study, only the BMI was taken into account; bioimpedance measurements were not performed. Protein deficiency in the group of patients with sarcopenia may be of an alimentary nature in the sense that it is associated with an unbalanced diet due to excessive intake of carbohydrates and fats; it could also be of an endogenous origin due to the predominance of catabolic processes, due to the activation of proinflammatory markers. The absence of differences in fibroblast growth factor in the main and control groups is explained by the multifocal property of this metabolic regulator in middle-aged and elderly people; its increase is probably associated with a compensatory decrease in insulin resistance, as well as correction of dyslipidemia due to activation of lipolysis, b-oxidation in the liver [28]. In a study conducted in South Korea in elderly people aged 70 to 84 years, a decrease in plasma FGF21 levels was observed in participants with decreased muscle strength than in participants with normal muscle strength. At the same time, there was an increase in FGF21 in the group with decreased muscle strength than in the control group, which confirms our results [32]. This may be due to the fact that the participants in our study were middle-aged.

When conducting a review of the literature review, there are not enough publications on the study of nitrates in sarcopenia, but it was found that the decreased expression of inhibitor of nitric oxide synthase (iNOS) in the early stage of myocyte damage leads to disruption of satellite cell accumulation and muscle regeneration [33]. It can be assumed that their increase in the main group manifests itself as a response to fibrosis, since nitric oxide acts as a neurotransmitter, improving blood circulation, as well as cellular respiration processes [34]. Indeed, low nitrate levels have been reported in older adults, and dietary expansion showed the potential benefit of a nitrate-rich diet on muscle strength and physical function in a large cohort of older women [35]. Among the supplements tested, nitrates have received ample evidence to support their acute beneficial effects on muscle strength [36]. In addition, vasodilators increase levels of vitamin D, which is known to maintain mineral homeostasis, as well as improve muscle function [34]. There is great interest in assessing muscle mass and strength and nitric oxide levels in patients taking vitamin D supplements. A placebo-controlled study conducted in 2020 found improvements in grip strength, decreased time to get up from stool, and decreased body fat in the main group taking supplements for 12 weeks [37].

Currently, malondialdehyde is considered to be a marker of oxidative stress, which is one of the links in the pathogenesis of sarcopenia [36]. Previously, reliable data on its increase in coronary heart disease, stroke, and liver damage were obtained [37]. Perhaps their increase is concomitantly associated with excess body weight, since there is evidence of increased malondialdehyde in persons with sarcopenic obesity [38], and in our work, we observed excess body weight in both the main and the control groups. As a result, changes in high-density lipoprotein levels and malondialdehyde–low-density-lipoprotein ratios were significantly and independently associated with changes in muscle strength [39]. Antioxidants (dark chocolate and astaxanthin) were found to reduce oxidative LDL and malondialdehyde levels for a month and increase nitric oxide levels, helping prevent muscle mass loss [40].

It was found that combined aerobic and strength training during body weight loss can reduce the levels of proinflammatory markers IL-6 and TNF-ɑ, due to a decrease in visceral fat and improved protein synthesis and myocyte cell membrane strength [41]. Extended studies are needed to further investigate the mechanisms by which physical activity affects chronic inflammatory processes and pro-inflammatory markers.

The pathophysiology of sarcopenia resembles that of other prototypical geriatric conditions for which a multifactorial etiology is at play, involving many of the biological hallmarks of aging (i.e., genomic and epigenetic instability, loss of proteostasis, mitochondrial dysfunction, telomere shortening, dysregulated nutrient signaling, stem-cell exhaustion, cellular senescence, and altered intercellular signaling). The currently available biomarkers of biological aging might not capture the multitude of intrinsic and extrinsic factors underlying the decline in physical function that characterizes sarcopenia.

The sample size was a limitation of the study, but material is being recruited for an epidemiologic evaluation of the potential of bioimpedance and metabolic markers in the early preclinical screening of probable sarcopenia and will be expanded. In addition, an older cohort is needed in the supplement to evaluate the content of predictive markers in confirmed sarcopenia and to provide a diagnostic algorithm; recruitment is ongoing and will be considered in future publications.

## 5. Conclusions

Anxiety, depression, and quality-of-life assessments show no change in middle-aged patients. Thus, for the primary screening of muscle mass reduction, it is objective to use bioimpedancemetry и dynamometry. These techniques will allow for the timely identification of patients from the risk group, which will allow early preventive and rehabilitative measures to be carried out. In addition, the use of bioimpedance as a method of assessing the dynamics of weight loss in overweight patients will make it possible to select a training program aimed at preserving muscle mass. Evaluation of the ratio of protein, minerals, and fluids in the body may be relevant for the development of an individual nutrition program. Early screening of the disease at the presarcopenia stage and preventive measures will reduce the number of patients with severe forms of sarcopenia, thereby increasing the quality of life and life expectancy of older people.

Because of the current high prevalence of carbohydrate metabolism disorders (type 2 diabetes mellitus and obesity), further study of the role of metabolic markers in sarcopenic obese individuals in a broader population is required. Further results on the study of metabolic markers may clarify new pathogenetic features associated with the development of sarcopenia. This will allow for the identification of new points for drug exposure, as well as the confirmation of statistical significance in a wider population, along with the use a panel of biomarkers for the predictive diagnosis of sarcopenia. I=The implementation of a multivariate predictive diagnostic methodology for sarcopenia will provide a way to stratify the risk of muscle mass loss, facilitate identification of a deteriorating condition, and provide monitoring of treatment effectiveness. In addition, such a marker as nitrates was not previously underestimated in the cross-section of decrease and loss of muscle mass and strength, which requires more in-depth study and will, in the future, probably open new fundamental ways of development of sarcopenia in middle age. Nitrates should be studied in more depth not only from the standpoint of sarcopenia, but also for predictive diagnosis in middle-aged individuals.

## Figures and Tables

**Figure 1 jpm-12-01830-f001:**
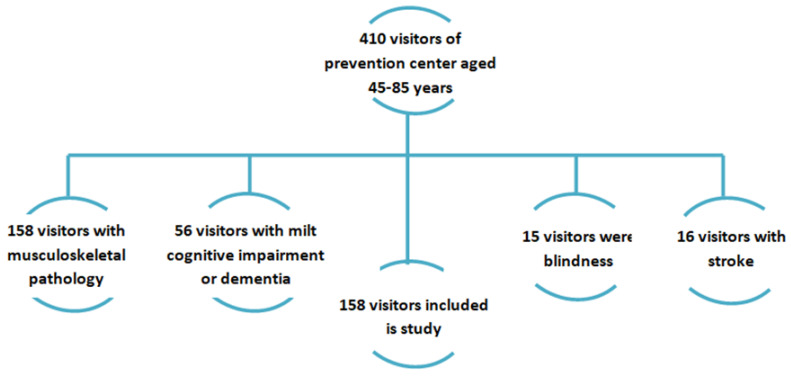
Recruitment protocol for the patients.

**Table 1 jpm-12-01830-t001:** Characteristics of included patients stratified by gender.

Parameters	Women		*p*	Men		*p*
	Main Group (*n* = 118)	Control Group (*n* = 28)		Main Group (*n* = 47)	Control Group (*n* = 28)	
Age, years	54 [48–65]	55 [48–66]	0.453	53 [49–59]	52 [47–56.5]	0.534
Body mass index kg/m^2^	28.5 [25.1–31.8]	28.5 [25.1–31.8]	0.285	28.5 [25.1–31.8]	28.5 [25.1–31.8]	0.285
Waist-to-hip ratio	0.97 [0.93–1.01]	0.96 [0.9–1.02]	0.435	0.94 [0.8–1.02]	0.95 [0.8–1.05]	0.435
Right hand, kg	24 [19–31]	40 [38–53]	0.001	17 [15–24]	35 [27–39]	0.001
Left hand, kg	23 [19–29]	38 [33–51]	0.001	15 [13–22]	33 [25–36]	0.001
Gait speed, m/s	0.95 [0.8–1.25]	1.15 [0.8–1.4]	<0.001	1.0 [0.9–1.3]	1.2 [0.9–1.5]	<0.001
Glycemia, mmol/L	4.9 [4.6–5.3]	4.95 [4.6–5.3]	0.834	4.7 [4.3–5.2]	4.5 [4.2–5.0]	0.745

**Table 2 jpm-12-01830-t002:** Parameters of neuropsychological testing in the main and control groups.

Options	Main Group	Control Group	*p*
SarQol	80.54 [67.89–86.895]	78.03 [63.81–89.19]	0.833
HADS (anaxiety)	4 [2–7]	4 [2–6]	0.108
HADS (depression)	4 [2–6]	3 [1–5]	0.142
SF36, physical component	55 [49–59]	56 [50–60]	0.228
SF36, psychological component	45 [42–48]	46 [43–49]	0.499

**Table 3 jpm-12-01830-t003:** Bioimpedancemetry parameters in the main and control groups.

Parameters	Women		*p*	Men		*p*
	Main Group (*n* = 118)	Control Group (*n* = 28)		Main Group (*n* = 47)	Control Group (*n* = 28)	
Body fat mass (kg)	28 [25.1–36]	26.5 [20.65–35]	0.011	30 [26.1–38]	26.5 [20.65–35]	0.002
Body fat percentage (%)	35.9 [35.4–41.3]	33.7 [29.85–40.1]	0.0001	39.7 [35.6–44.8]	35.1 [29.7–40.5]	0.0001
Visceral fat area (cm^2^)	144.2 [128.3–176.1]	129.1 [92.2–169.25]	0.002	154.7 [128.3–198.4]	132.3 [95.3–183.5]	0.002
Skeletal muscle mass (kg^2^)	25.5 [23.4–27.9]	28.6 [22.9–33.3]	0.043	23.2 [21.4–27.4]	26.4 [22.5–34.1]	0.003
Appendicular muscle mass (m^2^)	6.7 [5.9–7.3]	7.8 [6.5–8.9]	<0.001	6.4 [5.85–7.1]	7.6 [6.3–8.5]	<0.001
Protein content	8.9 [8.1–10.0]	9.5 [8.25–11.6]	0.048	8.4 [7.5–9.4]	9.3 [8.25–11.6]	0.002
Minerals	3.1 [2.8–3.4]	3.9 [2.9–4.0]	0.03	3.0 [2.5–3.5]	3.6 [2.8–3.9]	0.044

**Table 4 jpm-12-01830-t004:** Distribution of metabolic markers in patients with and without sarcopenia.

Metabolites	Main Group (F = M = 47)	Control Group (F = M = 28)	*p*
Fibroblast growth factor 21 ng/L	263.8 [251.4–274.8]	267 [246.5–273.5]	0.80
Nitrates, mmol/L	0.21 [0.21–0.355]	0.105 [0.07–0.14]	0.05
Malondialdehyde (MDA), µmol/L	290.32 [260.97–290.32]	147.42 [78.06–216.77]	0.76

## Data Availability

Not applicable.

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
