# Peer review of "Markers for the Prediction of Probably Sarcopenia in Middle-Aged Individuals"

_jpm, 2022, doi:10.3390/jpm12111830_

Round 1
Reviewer 1 Report (Previous Reviewer 2)
Dear authors,
The paper submitted by Samoilova et al. to the prestigious Journal of Personalized Medicine approaches a very interesting and contemporary topic regarding the predictive. markers of sarcopenia in middle-aged individuals.
I hope that my remarks and suggestion will be useful in order to increase the quality of the manusctipt.
1. Line 23 -Abstract- a more comprehensive/punctual conclusion is needed.
2. Line 28 – requires citation. Is the same source as the one from line 29?
3. Line 90 – a blank informed consent should be sent to the assistant editor.
4. Line 109 – For a better understanding i would suggest to increase the quality of Fig. 1
5. Line 112 – Can you please elaborate on the testing method?
6. Lines 294-298 – Did you check the possibility to asses the mentions markers like ILs and TNF at salivary level? Did you find any relevant literature on this method in relation with sarcopenia?
7. Lines 313-333 – Conclusions section – From my point of view, the conclusions section should be rephrased in a much synthetic manner focusing on the practical/clinical impact or relevance of your findings in order to improve cuurent protocols. In the same time part of the pieces of information included in this section should be integrated in the Disccusions.
Please receive my best regards!
Author Response
Dear reviewer, thank you for the recommendations!
- Line 23 -Abstract- a more comprehensive/punctual conclusion is needed.
Answer: Thanks for the comment, expanded and added to the abstract
Sarcopenia is a condition characterized by a progressive loss of muscle mass, strength, and function, that reduced quality of life. The aim of the study was to analyze the significance of pro-inflammatory markers in the prognostic diagnosis of sarcopenia. The participants were divided into two groups - the main group of 146 people and the control - 75 people. The complex of examinations included: neuropsychological testing (Hospital Anxiety and Depression Scale (HADS), quality of life questionnaire for patients with sarcopenia (SarQoL), short health assessment form (MOS SF-36)), a 6 m walking speed test, manual dynamometry, bioimpedancemetry, metabolic markers (nitrates, fibroblast growth factor-21, malondialdehyde). When analyzing metabolic markers in the main group, a twofold increase in nitrates in the main group was recordedIn a subsequent analysis adjusted for multiple variables, there was a negative association of nitrate levels for weak grip strength, appendicular muscle mass. Additional analysis revealed: the complaint of pain in the lower extremities was more frequent in patients of the main group, as well as constipation, pathology of thyroid gland and they were more frequently diagnosed with arterial hypertension. At the same time, patients from the main group more frequently took vitamin D. When conducting body composition, the main group recorded higher weight, visceral fat content, as well as a decrease in appendicular and skeletal muscle mass, these changes were accompanied by a decrease in protein and minerals. Among the markers that differed significantly were nitrates, and it was this that was associated with decreased muscle strength and appendicular mass, which may indicate both a possible mechanism and a possible predictive marker. The results of this study can be used to develop a screening method for diagnosing sarcopenia at the outpatient stage.
- Line 28 – requires citation. Is the same source as the one from line 29?
Answer: Thanks for the comment, we have included an additional link
- Line 90 – a blank informed consent should be sent to the assistant editor.
Answer: We attached the consent information earlier, but I duplicated it now.
- Line 109 – For a better understanding i would suggest to increase the quality of Fig. 1
Answer: We change the picture
- Line 112 – Can you please elaborate on the testing method?
Answer: we elaborate the testing methods
Dynamometer and walking speed test were used as primary testing to differentiate into groups. Patients were divided into groups according to the results of physical strength and function assessment, thus belonging to the group according to the classification features "probable sarcopenia". The protocol included measuring handgrip strength while taking maximum readings in two trials using both arms in isometric contraction with maximum force, the standard was sitting position with 90° elbow extension for the dynamometer. The diagnostic criteria for low handgrip muscle strength < 28.0 kg for men and < 18.0 kg for women [19]. To assess the functions, we measured the time taken to walk 6 meters at a normal pace from the start without slowing down, and took the average result of at least two trials using a handheld stopwatch as the recorded speed. A level ≤ 0.8 m/s to < 1.0 m/s, independent of floor, was used for the criteria.
- Lines 294-298 – Did you check the possibility to asses the mentions markers like ILs and TNF at salivary level? Did you find any relevant literature on this method in relation with sarcopenia?
Answer:Currently, our group has collected material in the form of saliva to evaluate interleukins and look for correlation, no such publications on sarcopenia have been found, we are searching.
- Lines 313-333 – Conclusions section – From my point of view, the conclusions section should be rephrased in a much synthetic manner focusing on the practical/clinical impact or relevance of your findings in order to improve cuurent protocols. In the same time part of the pieces of information included in this section should be integrated in the Disccusions.
Answer:
We have adjusted these sections in the manner suggested
Thank you for such an important comment and review!
Reviewer 2 Report (Previous Reviewer 1)
It is a very informative manuscript.
- The question addressed in this manuscript is important and relevant, particularly to researchers on sarcopenia.
- The review quoted published data which I suggested that authors should not rely on Tabled information only. They should write up a summarizing concluding section.
- Readers could be puzzled on the tabled data. Authors could help readers with a summarizing section on the overall results given by the different groups of investigators.
Author Response
Dear reviewer, thank you for the recommendations!
- The review quoted published data which I suggested that authors should not rely on Tabled information only. They should write up a summarizing concluding section.
Answer: Thanks for the comment, this section has been expanded.
- Readers could be puzzled on the tabled data. Authors could help readers with a summarizing section on the overall results given by the different groups of investigators.
Answer:We included the information in the manuscript
Thank you for taking the time to read our manuscript
This manuscript is a resubmission of an earlier submission. The following is a list of the peer review reports and author responses from that submission.
Round 1
Reviewer 1 Report
It is a very informative manuscript. The section on Results relies totally on giving tables, which are useful but cumbersome. Would suggest authors to give a short summing up paragraph on the overall results to help readers to correlate between the different groups of investigation.
Author Response
Thank you for such a valuable comment.
At the end of the section we summarized the results obtained, it will help readers to understand the essence of the study more easily.
We also gave an introductory sentence at the beginning to bring readers up to speed.
Reviewer 2 Report
Dear Authors,
Thank you very much for submitting your manuscript to JPM. The paper approaches an interesting toăpic and I hope that my remarks will be useful in order to increase its' quality.
Line 83 - The authors should provide a blank informed consent to the editorial team. If you already did that please ignore the remark.
Line 87 - please elaborate on how the exclusion criteria were set.
Line 173 - "Discussion" section requires to be more extension. Please consult more recent literature related to the topic and report it in comparison to your results.
Line 246 - please elaborate on the limitations of your study.
Line 249 - From my point of view, the "Conclusions" section should be rephrased in a much synthetic way, in order to emphasise the future clinical/practical impact of your research.
Best regards!
Author Response
Thank you for such a comprehensive comment.
Line 83 - The authors should provide a blank informed consent to the editorial team. If you already did that please ignore the remark.
Thank you, we sent (attached) an additional blank informed consent.
Line 87 - please elaborate on how the exclusion criteria were set.
Of the many criteria, those that could affect the results of the study have been chosen, meaning that it is they, not sarcopenia, that will determine the results obtained even as early as the first stage of randomization, which would violate the purity of the experiment.
Line 173 - "Discussion" section requires to be more extension. Please consult more recent literature related to the topic and report it in comparison to your results.
Thank you, indeed not all aspects were taken into account, so this section was expanded.
Line 246 - please elaborate on the limitations of your study.
Thank you, you've expanded the paragraph of limitations: Sample size is a limitation of the study, but material is being recruited for an epidemiologic evaluation of the potential of bioimpedance and metabolic markers in the early preclinical screening of probable sarcopenia and will be expanded. In addition, an older cohort is needed in the supplement to evaluate the content of predictive markers in confirmed sarcopenia and to provide a diagnostic algorithm; recruitment is ongoing and will be considered in future publications.
Line 249 - From my point of view, the "Conclusions" section should be rephrased in a much synthetic way, in order to emphasise the future clinical/practical impact of your research.
Thank you for the recommendation, we have expanded this section to include what we think are important points.
As for the practical part, preclinical diagnostics will help to carry out early interventions, in this case it is dietary prerequisites, dietary adjustments based on the indicators of body composition on an individual basis, that is personalized. In addition, such a marker as nitrates was not previously underestimated in the cross-section of decrease and loss of muscle mass and strength, which requires more in-depth study, which in the future will probably open new fundamental ways of development of sarcopenia in middle age.
Reviewer 3 Report
Minor misprints through the manuscript that should be corrected.
Why is blindness a characteristic use to exclude patients? Properly justify your response.
Figure 1 should be enhanced (>600 dpi) as one cannot zoom in to properly read the information.
There I no justification on the 10 kg difference to measure handgrip between men and women. A reference should be added.
Windows versions are not necessary to be described in the methods, the version of the SPSS is enough.
Authors describe their methods but do not justify the values that they applied for their thresholds, which should be discussed within this section.
Results should be redone completely, as it presented in this version it looks like only the mean values was computed in the data when authors clearly stated the following:
“Methods of descriptive statistics were: mean value and standard deviation - for normally distributed data; quartiles- for non-normally distributed data. Qualitative binary signs were presented in the form of relative frequency (%) and its 95% confidence interval. Testing of statistical hypotheses of normally distributed quantitative parameters was performed using the following parametric criteria: Student's test for paired comparison (when assessing independent samples), Student's test for dependent data (dependent), analysis of variance for multiple comparisons. Correlation analysis was assessed using Pearson's criterion. Nonparametric criteria were used to test hypotheses for non-normally distributed quantitative parameters: Mann-Whitney (independent samples) and Wilcoxon (dependent samples), Kruskal Wallace criteria for paired comparison. For correlation analysis - nonparametric Spearman criterion. To assess reliability of differences in qualitative characteristics, we used conjugation tables with the calculation of χ² (chi-square). The null hypothesis was rejected at the level of statistical significance p <0.05.”
None of these statistics were presented in the results section.
Mathematically a set of date with closed values should be presented as [x,y] not [x:y], and the mean values should be in its own column.
Discussion and conclusions of the work do not provide a proper framework for the prediction of ‘probably’ sarcopenia in middle aged individuals. If the information is still hidden in their results, authors should provide a properly present all their findings in the Results section of the manuscript.
Authors did not properly used the template of MDPI.
Introduction needs to provide a better background for their work, authors should make a better review of the current literature regarding their work.
Author Response
Why is blindness a characteristic use to exclude patients? Properly justify your response.
Thank you for your comment. Vital functions, including blindness, can affect physical activity and muscle mass, so we adopted this exclusion criterion in this study.
Figure 1 should be enhanced (>600 dpi) as one cannot zoom in to properly read the information.
Thanks for the comment, converted to the recommended format
Windows versions are not necessary to be described in the methods, the version of the SPSS is enough.
Thanks for the comment, removed unnecessary information.
Authors describe their methods but do not justify the values that they applied for their thresholds, which should be discussed within this section.
Thank you, we clarified the thresholds
Results should be redone completely, as it presented in this version it looks like only the mean values was computed. None of these statistics were presented in the results section.
This section has been corrected for the results given in this article.
Mathematically a set of date with closed values should be presented as [x,y] not [x:y], and the mean values should be in its own column.
Thanks for the correction.
Discussion and conclusions of the work do not provide a proper framework for the prediction of ‘probably’ sarcopenia in middle aged individuals. If the information is still hidden in their results, authors should provide a properly present all their findings in the Results section of the manuscript.
Thank you, we have expanded these sections
Authors did not properly used the template of MDPI.
We changed it to the right one template of MDPI.
Round 2
Reviewer 3 Report
Discussion and conclusions of the work do not provide a proper framework for the prediction of ‘probably’ sarcopenia in middle aged individuals. If the information is still hidden in their results, authors should provide a properly present all their findings in the Results section of the manuscript.
Author Response
Dear Reviewer!
Thank you for your comments, we have corrected the introduction and the discussion, as well as the references to the list of references.
Nothing new has been added on method description issues, they are taken from the clinical guidelines and conducted according to the standard. What should be added to this part, if possible clarify.
Currently, the study is ongoing and expanded its territories, but it is not yet possible to include these data in the results of the work.
Round 3
Reviewer 3 Report
No further comments.